# Effects of Solution Temperature on Tensile Properties of a High γ′ Volume Fraction P/M Superalloy

**DOI:** 10.3390/ma15165528

**Published:** 2022-08-11

**Authors:** Jian Jia, Yiwen Zhang, Yu Tao, Ting Yan, Hongyan Ji

**Affiliations:** 1High Temperature Materials Research Institute, Central Iron and Steel Research Institute, Beijing 100081, China; 2Gaona Aero Material Co., Ltd., Beijing 100081, China

**Keywords:** superalloy, powder metallurgy, solution temperature, precipitation strengthening

## Abstract

To meet the pressing needs concerning the optimization of the performance of powder metallurgy (P/M) superalloys for turbine disc applications, the effects of solution temperature on a novel high γ′ volume fraction P/M superalloy FGH 4107 were investigated. The results indicated that the size of the γ′ precipitates decreased dramatically as the solution temperature increased from 1160 to 1200 °C. Theoretical calculations showed that the precipitation strengthening played a dominant role in enhancing the strength of the high γ′ volume fraction P/M superalloy, and a higher solution temperature was beneficial for the modification of the γ′ phase distribution during the following cooling and aging process.

## 1. Introduction

With excellent microstructural homogeneity and comprehensive properties at elevated temperatures, powder metallurgy (P/M) superalloys are deemed to be ideal materials to manufacture turbine disks for advanced aircraft engines [1,2,3,4,5]. As turbine disks are subject to complex centrifugal force during service, adequate tensile strength in the P/M superalloy is crucial to guarantee the safety of aircraft. Specifically, a yield strength (YS) exceeding 1000 MPa and ultimate tensile strength (UTS) greater than 1200 MPa at elevated temperature are commonly considered essential for disc superalloys [6,7].

Composition and process optimization are two major methods that have been used to improve the tensile strength of P/M superalloys in recent decades. In terms of the composition, the refractory alloying elements, including Mo and W, yield a solid-solution-strengthening effect [8,9]. The γ′ phase in the A3B type (A = Ni, Co; B = Ti, Al, Nb, Ta, etc.) is a typical phase in superalloys, and the γ′ phase-forming elements, such as Ti, Nb, Al and Ta, have a precipitation-strengthening effect [8,10]. Cr is important to enhance oxidation resistance [11]. Co can reduce stacking fault energy and enhance creep resistance [12,13]. Minor elements including Zr, B and Hf are also indispensable for the tailoring of mechanical properties by purifying or stabilizing grain boundaries [14,15,16,17,18]. Strengthening via solid solution strengthening and precipitation plays a dominant role in improving alloy strength, while excessive refractory elements lead to the formation of topologically close-packed (TCP) phases, which are detrimental to mechanical properties [19,20,21]. Therefore, to meet the requirement for the increased service temperature of disks, there is a trend of adding more γ′-forming elements to enhance the precipitation strengthening for P/M superalloys, and the γ′ fraction may approach nearly 60% [22,23,24]. However, increasing the γ′ fraction makes it difficult to perform thermomechanical manufacturing successfully [25]. In general, the P/M route involves several processes; specifically, powder preparation, hot isostatic pressing, extrusion and forging, followed by heat treatment [26]. To avoid cracking during extensive plastic formation, hot isostatic pressing without ensuing extrusion or forging, known as “as-HIP”, is applied for high γ′ fraction P/M superalloys, such as EP741NP [27]. Regarding the process optimization, modifying the heat treatment parameters is an effective way to improve tensile properties. By modeling the relationship between the microstructure and yield strength of the P/M nickel-base superalloy RR1000, Collins and Stone [1] found that the optimal yield strength properties at 600–700 °C could be obtained when γ′ precipitates were in the range of 34–57 nm, which has thereafter guided optimizations of the solution and aging to obtain maximized high-temperature strength.

To satisfy the need to boost the high-temperature performance of P/M superalloys for turbine disc applications, a novel high γ′ volume fraction P/M superalloy was designed and manufactured via as-HIP. How to further promote its mechanical properties via modification of the heat treatment process remains to be investigated. In this work, the as-HIPed high γ′ volume fraction P/M superalloy was prepared by changing the solution temperature and the effects of the solution treatment on the microstructure and tensile properties were studied systematically, which may give hints about how to obtain the optimum process route for high γ′ volume fraction P/M superalloys.

## 2. Materials and Methods

As indicated in Figure 1, to prepare the high γ′ volume fraction P/M superalloy, the master alloys were first produced through vacuum induction melting (VIM) of high-purity raw materials (>99.99%). The raw materials were supplied by Jinchuan Group International Resources Co. Ltd. (Gansu, China), Ningxia Orient Tantalum Industry Co., Ltd. (Ningxia, China) and others. Then, prealloy powders with sizes of 0–75 μm were prepared using electrode induction gas atomization (EIGA, Seller/Manufacturer: ALD Vacuum Technologies, Hanau, Germany) and sieving. Before HIP at 1200 °C and 130 MPa for 4 h, the prealloy powders were loaded into a steel container and degassed, and the container was sealed by welding. After removing the outer steel, a billet of as-HIPed P/M superalloy with a diameter of 82 mm and height of 135 mm was obtained, and its detected composition is presented in Table 1 based on the GB/T 222 standard.

To investigate the effects of heat treatment on the tensile properties, the billet was divided into three sections that were then solutioned at 1160, 1180 and 1200 °C, respectively, for 1.5 h, during which the average heating rate and cooling rate were about 100 °C/h and 90 °C/min, respectively. Next, these sections were immediately quenched in a mixed salt bath (SA) consisting of NaCl, BaCl_2_ and CaCl_2_, and then subjected to a two-step aging treatment; i.e., 871 °C aging for 4 h with air cooling and then 760 °C aging for 16 h with air cooling. Four stages were thus involved in the preparation of the FGH4107, as listed in Table 2.

The tensile testing was conducted at ambient temperature, 650 and 700 °C, in accordance with the GB/T 228.2-2015 standard. The fractographies and γ′ morphologies were observed with a Quanta 650 FEG field emission scanning electron microscope (FE-SEM), which was equipped with an electron backscattering diffraction (EBSD) detector. Moreover, for TEM observation, slices with a thickness of about 50 μm and diameter of 3 mm were cut from samples and then twin-jet electropolished in reagent of 90 vol.% C_2_H_5_OH + 10 vol.% HClO_4_ at −25 °C and 20 V. A Titan G2 60–300 high-angle annular detector dark-field scanning transmission electron microscope (HAADF-STEM) was used to obtain the local element distribution in the P/M superalloy. To determine the electron back-scattered diffraction (EBSD) and determine the grain size, these alloys were polished with abrasive papers and vibration polishing. Moreover, thermodynamic equilibrium calculations were adopted to calculate the phase diagram of FGH4107 in the software JMatPro 6.0 (Version 6.0, Sente Software Ltd., Guildford, UK).

## 3. Results

### 3.1. Initial Microstructure

Figure 2 shows the FGH4107 EBSD inverse pole figures (IPFs), which indicate that grains in these specimens were uniform and fine. In comparison, the average grain size of HT-3 was greater than that of the other samples. Specifically, the average grain size of the as-HIPed FGH4107 was 23.90 μm, and the average values of heat-treated HT-1, HT-2 and HT-3 samples were 23.15, 24.33 and 30.85 μm, respectively. As indicated in Figure 3, the frequency of large grains in HT-3 with equivalent grain sizes over 40 μm was higher than in As-HIP, HT-1 and HT-2.

The morphologies of FGH4107 after hot isostatic pressing are shown in Figure 4. Large, irregular primary γ′ precipitates with equivalent diameters of nearly 3 μm were detected along the grain boundaries. The secondary γ′ precipitates within the grains had subangular or cubic shapes. A small amount of tiny tertiary γ′ precipitates dispersed among the large γ′ precipitates was also observed.

The solution and aging treatment had a strong influence on the morphologies and distribution of the γ′ precipitates. As shown in Figure 5, the primary γ′ precipitates located along the grain boundary became smaller as the solution temperature increased from 1160 to 1200 °C. After heat treatment, the secondary γ′ precipitates in these P/M superalloy samples presented a typical octocube morphology. In contrast, the edges of the secondary γ′ precipitates in the samples solutioned at 1160 and 1180 °C had many bulges, while the edges of the secondary γ′ precipitates in the sample solutioned at 1200 °C were smooth. As the solution temperature increased, the size of the γ′ precipitates obviously decreased.

As indicated in Figure 6, the solution temperature had a non-obvious influence on the elemental distribution in the γ′ phase and γ matrix of FGH4107. Al and Ti are strong γ′ phase formation elements and they are rich in γ′ phase, while Cr and Co are prone to segregate in the γ matrix.

The carbides embedded in grains were all found in the aged FGH4107. As illustrated in Figure 7, these carbides were rich in Ti, Nb and Ta and, using the selected area electron diffraction (SAED) results, they were identified as MC-type carbides.

### 3.2. Tensile Properties

The tensile properties of the P/M superalloy FGH4107 solutioned at different temperatures are illustrated in Figure 8. In general, alloys solutioned at higher temperatures have superior yield and ultimate tensile strength. At room temperate, the yield strength increased by 52 MPa, and the elongation was also slightly increased when the solution temperature increased from 1160 to 1200 °C, while the effect of the solution temperature on the ultimate tensile strength at ambient temperature was not obvious. At 650 °C, with the increase in the solution temperature, the yield and ultimate tensile strength fluctuated slightly, while the ductility dropped from 17.5 to 15.0%. Moreover, increasing the solution temperature to 700 °C significantly enhanced the strength of P/M superalloy FGH4107, as the yield strength and ultimate tensile strength improved by 50 MPa and 150 MPa, respectively, and the elongation dropped from 18% to 16% correspondingly.

Fracture morphologies of the FGH4107 ruptured at ambient temperature are illustrated in Figure 9. Three regions—i.e., crack nucleation, crack propagation and final rupture zones—could clearly be observed. Specifically, the cracks of these three samples began at the surface, and ductile intergranular tears and quasi-cleavage facets could also be observed, indicating ductile fracture characteristics. The crack propagation zones occupied the largest area on the fracture surfaces, and intergranular cracks, ductile intergranular tears and quasi-cleavage facets had dominant roles there.

As shown in Figure 10, when the tensile temperature increased to 650 °C, the area of the final rupture zones grew. In addition, as typical brittle fracture characteristics, more quasi-cleavage facets appeared at crack nucleation zones and crack propagation zones for all these three FGH4107 samples, contributing to the decrease in the elongation after fracture at 650 °C.

As shown in Figure 11, at 700 °C, cleavage facets still played dominant roles in the crack nucleation regions of FGH4107, while more ductile intergranular tears appeared in the crack propagation zones, especially for the FGH4107 samples solutioned at 1180 and 1200 °C, which contributed to the improvement of the tensile ductility.

## 4. Discussion

For polycrystalline superalloys, solid solution strengthening, grain boundaries and precipitation are the three major factors contributing to the general strength [3].

Regarding the solid solution strengthening complemented with aging at 760 °C for 16 h, the aging processes for these FGH4107 samples with the same compositions were identical; hence, the element distribution in the different phases after long-term aging can be regarded as the same. Therefore, the differences in the solid solution strengthening for these samples can be neglected. The compositions of both the γ matrix and γ′ phase at 760 °C were obtained using the equilibrium phase calculation, as listed in Table 3.

$S_{i}^{\gamma }$, the solid solution strength of the γ matrix contributed by the element *i*, was calculated as [28,29]:(1)$$S_{i}^{\gamma }=\beta_{i}^{\gamma }{\left(x_{i}^{\gamma }\right)}^{1/2}$$
where $x_{i}^{\gamma }$ is the mole fraction of element *i* in the *γ* matrix, and $\beta_{i}^{\gamma }$ represents the strengthening coefficient of Ni-base alloys, as summarized in Table 4 [9,28,29].

Therefore, the yield strength contributed by the solid solution of the *γ* matrix $\sigma_{\mathrm{sss}}^{\gamma }$ could be estimated by [28,30]:(2)$$\sigma_{\mathrm{sss}}^{\gamma }=f_{\gamma }{\left[\sum_{i}{\left(S_{i}^{\gamma }\right)}^{2}\right]}^{1/2}$$
where *f*_γ_ is the *γ* matrix volume fraction, which equals 1—*f*_γ′_; and *f*_γ′_ is the total volume fraction of γ′, which was determined to be 58.55%. Therefore, the value of for the FGH4107 superalloy was calculated to be 229.80 MPa.

According to the Hall–Petch law, the grain boundary strengthening *σ*_HP_ can be determined by [29]:(3)$$\sigma_{\mathrm{HP}}=\frac{k_{\mathrm{HP}}}{\sqrt{D}}$$
wherein *k*_HP_ is a constant, which is 710 MPa·μm^−1/2^ for superalloys [31], and *D* represents the average grain size. Therefore, the *σ*_HP_ values of HT-1, HT-2 and HT-3 were estimated to be 147.56 MPa, 143.94 MPa and 127.83 MPa, respectively.

In addition, the size and volume fraction strongly affect the precipitation strengthening. The volume fractions of the primary and secondary γ′ particles, and the average sizes of the primary, secondary and tertiary γ′ particles, were determined in ImageJ software. The volume fractions of the tertiary γ′ particles were calculated by assuming the sum of the volume fractions of the primary, secondary and tertiary γ′ phase to be equal to the equilibrium volume fraction of the γ′ phase at 760 °C; i.e., 58.55%. The characteristics of the γ′ phase in P/M superalloy FGH4107 are summarized in Table 5.

To determine the multimodal particle size distribution (PSD), the precipitation strengthening *σ*_P_ was evaluated as follows [32]:(4)$$\sigma_{P}=M\sqrt{{\left(\tau_{p,\;\mathrm{pri}}\right)}^{2}+{\left(\tau_{p,\;\sec}\right)}^{2}+{\left(\tau_{p,\;\mathrm{ter}}\right)}^{2}}$$
where *τ*_p,pri_, *τ*_p,sec_ and *τ*_p,ter_ correspond to the critical resolved shear stress (CRSS) of the primary, secondary and tertiary γ′ precipitates, and the Taylor factor *M* was taken as 3.06. The deduction of *τ*_p_ can be found in a previous study [3]. Specifically, a united model combining dislocation-particle configurations in both weak and strong pair-coupling cases was adopted [28,31]:For the leading dislocation:   *τ*_p_*bΛ*_1_ + *F*_R_*Λ*_1_ − *γ*_APB_*l*_1_ + 2*τ*_p_*br*= 0 
And for the trailing dislocation:    *τ*_p_*bΛ*_2_ − *F*_R_*Λ*_2_ = 0(5)

After integrating the above equations, *τ*_p_ could be expressed by [28]:*τ*_p=_*γ*_APB_*l*_1_/[2*b*(*Λ*_1+_*r*)](6)
where *Λ*_1_ and *Λ*_2_ are the distances of the γ′ precipitates sheared by the leading and trailing dislocations, and *γ*_APB_ represents the anti-phase boundary energy (APBE). According to the results of density functional theory calculations, the APBE of multicomponent Ni-base superalloys can be identified using the following equation [33]:(7)$$E_{\mathrm{APB}}=E_{{}_{\mathrm{APB}}}^{0}+\sum_{i=1}^{n}k_{i}x_{i}^{\gamma \prime }$$
where $x_{i}^{\gamma \prime }$ is the atom fraction of element *i* in γ′, as shown in Table 3. The coefficients of the different elements *k*_i_ are presented in reference [33], and the APBE of Ni_3_Al $E_{{}_{\mathrm{APB}}}^{0}$ hs been measured as 195 mJ/m^2^ [34]. Therefore, the APBE value of FGH4107 was determined to be 368.8 mJ/m^2^. *F*_R_ is the dislocation pair force at unit length. Burgers vector *b* equals *a*_γ′_/$\sqrt{2}$ for 1/2 < 110 > dislocation in FCC alloys, and lattice constants of the γ′ phase *a*_γ′_ for the alloy can be estimated with the following equation [35]:(8)$$a_{\gamma \prime }=a_{Ni_{3}Al}+\sum V_{{}_{i}}^{\prime }x_{{}_{i}}^{\prime }$$
where $a_{Ni_{3}Al}$ indicates the lattice parameters of Ni_3_Al, $V_{{}_{i}}^{\prime }$ represents the Vegard coefficient of element *i* in Ni_3_Al and $x_{{}_{i}}^{\prime }$ represents the mole fraction of *i* in the γ′ phase. *l*_1_ represents the length of leading dislocation cutting into the precipitate, which can be calculated by [31]:(9)$$l_{1}=\left\{\begin{array}{l} 2r\;\;\;\;\;\;\;\;\;\;\;\;(r<r_{m}\;\mathrm{Weak}\;\mathrm{pair}-\mathrm{coupling})\\ 2\sqrt{r^{2}-{\left(r-r_{m}\right)}^{2}}\;\;\;(r\geq r_{m}\;\mathrm{Strong}\;\mathrm{pair}-\mathrm{coupling}) \end{array}\right.$$
where *r*_m_ is the critical value determining the weak or strong pair-coupling condition, and *r* is the radius of the γ′ particle [31]:(10)$$\;r_{m}=\frac{Gb^{2}}{\gamma_{\mathrm{APB}}}$$
where *G*, which was determined to be 87.32 GPa, is the shear module [31].

In addition, *Λ*_1_ can be calculated with the following equation [31]:(11)$$\Lambda_{1}=\max(\lambda_{1},\;L-l_{1})=\max(L{\left(\frac{T}{\gamma_{\mathrm{APB}}r}\right)}^{1/2},\;L-l_{1})$$
where the mean particle spacing *L* can be deduced as follows:(12)$$L={\left(\frac{2\pi }{3f}\right)}^{1/2}r$$
where *f* is the volume fraction of the particles in the matrix.

Combining the above equations, the values of *σ*_p_ for HT-1, HT-2 and HT-3 were estimated to be 740.50, 771.20 and 811.33 MPa. Therefore, the greatest contribution to the yield strength at ambient temperature was made by precipitation strengthening, and HT-3 had the highest *σ*_p_ because of its fine distribution of γ′ phase, as can be judged from Figure 12.

## 5. Conclusions

In summary, this work investigated the effects of solution temperature on the tensile properties of a high γ′ volume fraction P/M superalloy FGH4107. The following conclusions can be drawn:(1)The solution and aging treatment had a strong influence on the morphologies and distribution of γ′ precipitates. As the solution temperature increased from 1160 °C to 1200 °C, the size of the γ′ precipitates obviously decreased.(2)At 650 °C tensile temperature, more quasi-cleavage facets appeared in the crack nucleation and crack propagation zones, and elongation after fracture decreased. However, more ductile intergranular tears appeared in the crack propagation zones at 700 °C, contributing to the improvement of tensile ductility.(3)For the high γ′ volume fraction P/M superalloys, the strength contributed by precipitation strengthening was the largest among the three mechanisms, and HT-3 had the highest yield strength because of its fine distribution of γ′ phase caused by the high solution temperature.

## Figures and Tables

**Figure 1 materials-15-05528-f001:**
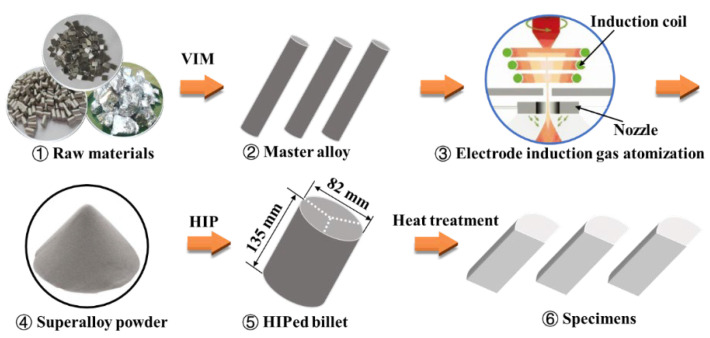
Schematic diagram of material preparation.

**Figure 2 materials-15-05528-f002:**
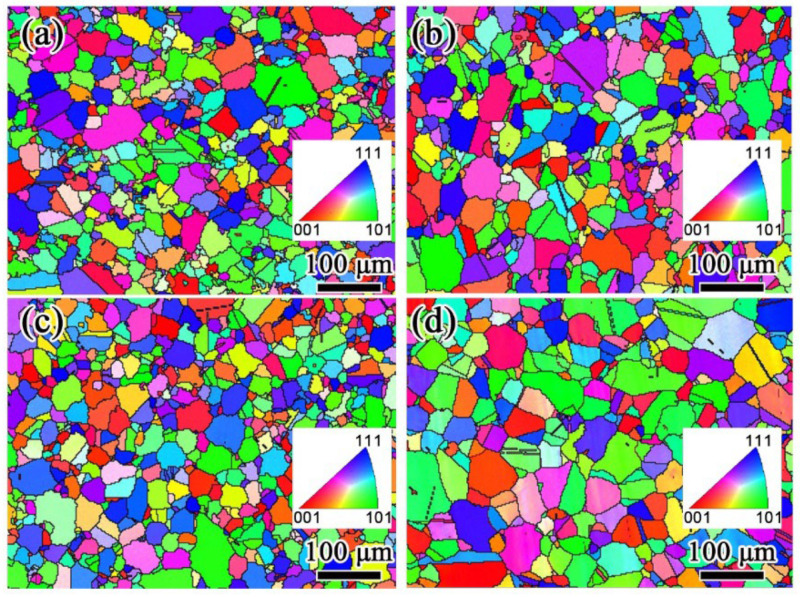
EBSD IPF images of FGH4107: (**a**) as-HIPed; (**b**) HT-1, (**c**) HT-2 and (**d**) HT-3.

**Figure 3 materials-15-05528-f003:**
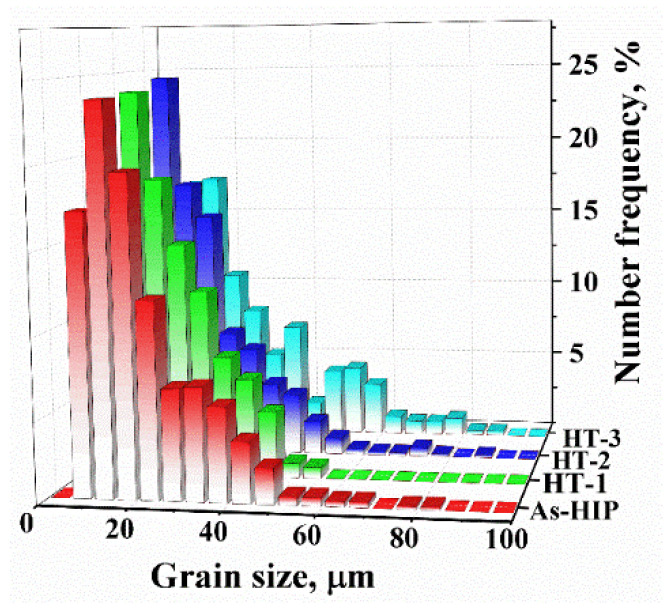
Grain size distribution for P/M superalloy FGH4107 specimens.

**Figure 4 materials-15-05528-f004:**
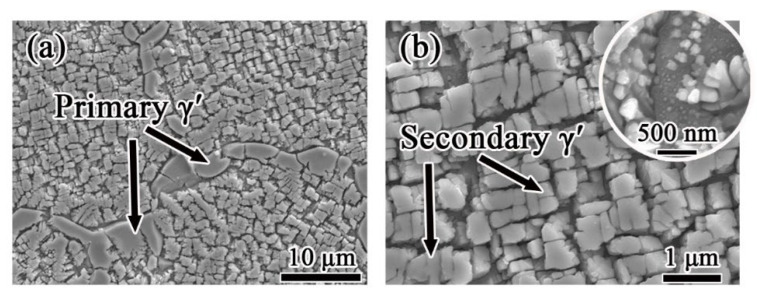
γ′ morphologies of FGH4107 after hot isostatic pressing: (**a**) primary γ′; (**b**) secondary γ′.

**Figure 5 materials-15-05528-f005:**
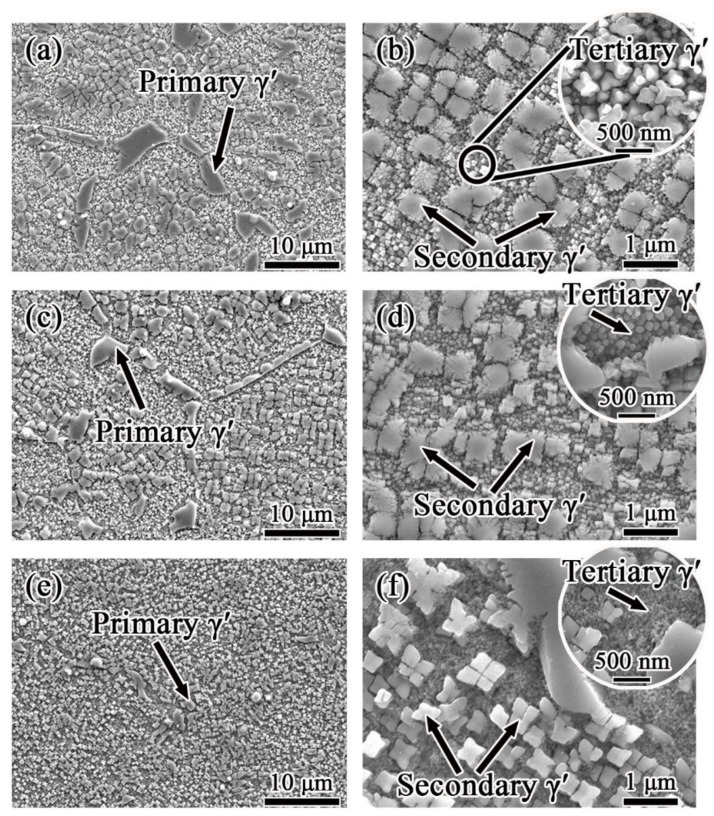
γ′ morphologies of FGH4107 solutioned at different temperatures: (**a**,**b**) 1160 °C; (**c**,**d**) 1180 °C; (**e**,**f**) 1200 °C.

**Figure 6 materials-15-05528-f006:**
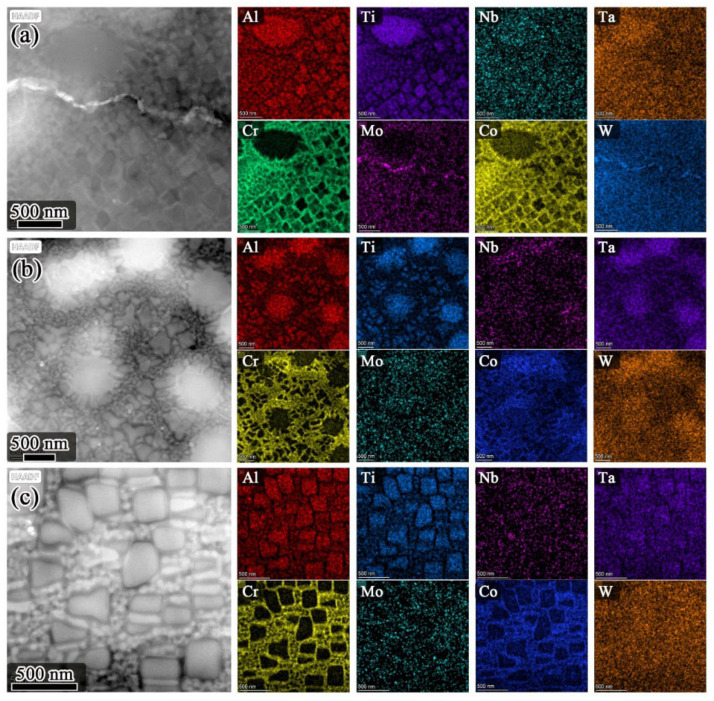
HADDF-STEM images and corresponding EDS mappings showing the elemental distribution in the γ′ phase and γ matrix of FGH4107 solutioned at different temperatures: (**a**) 1160 °C; (**b**) 1180 °C; (**c**) 1200 °C.

**Figure 7 materials-15-05528-f007:**
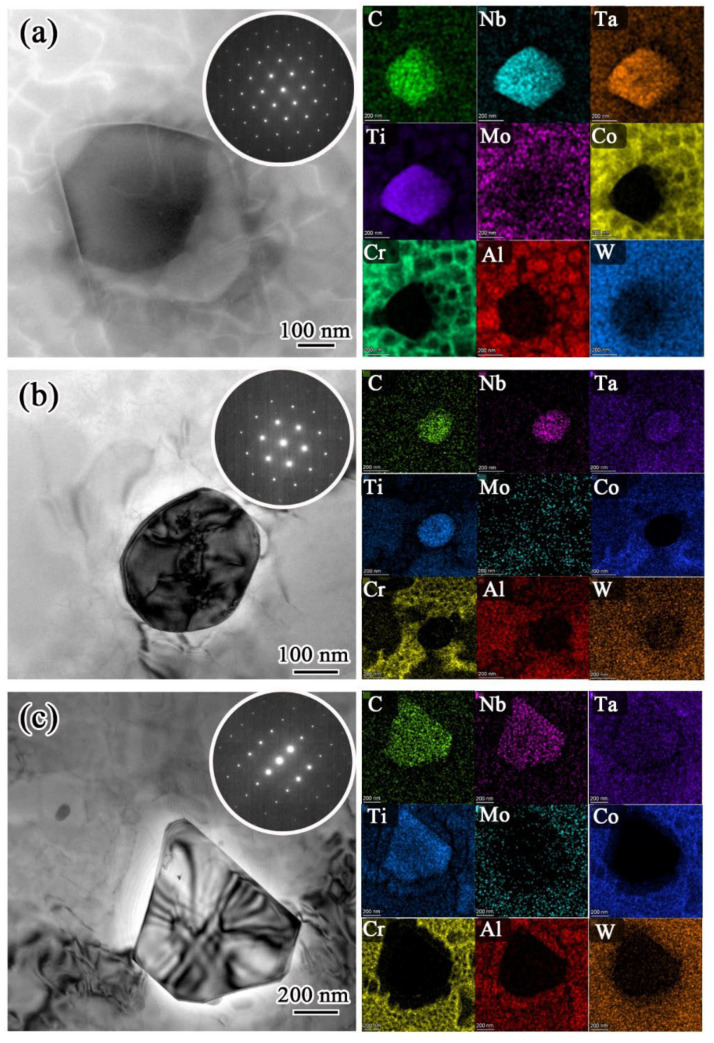
HADDF-STEM images and corresponding EDS mappings showing the morphologies and elemental distribution of carbides in FGH4107 solutioned at temperatures of: (**a**) 1160 °C; (**b**) 1180 °C; and (**c**) 1200 °C.

**Figure 8 materials-15-05528-f008:**
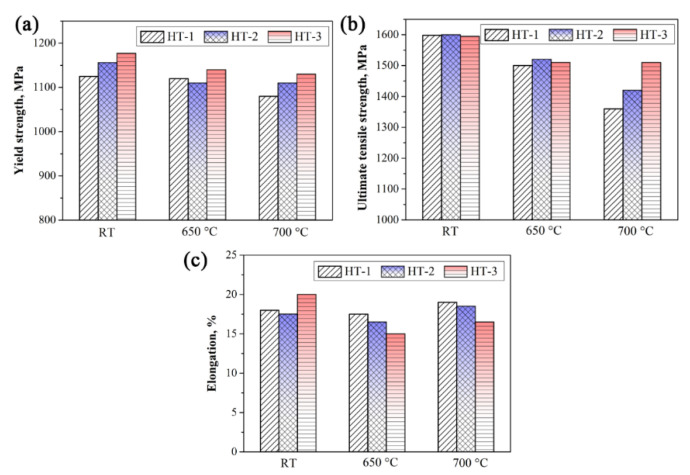
Tensile properties of the heat-treated P/M superalloy FGH4107: (**a**) yield strength; (**b**) ultimate tensile strength; (**c**) elongation.

**Figure 9 materials-15-05528-f009:**
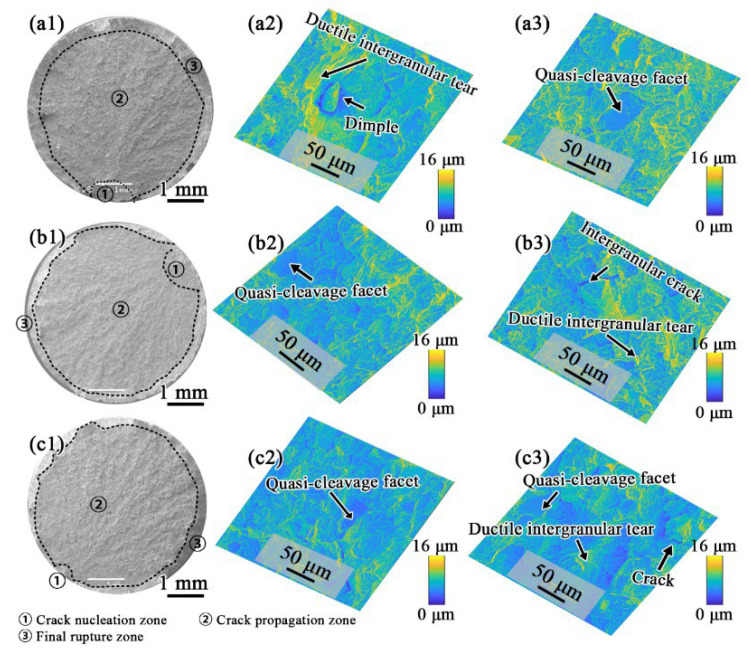
Fractographies of room temperature tensile specimens of (**a1**) HT-1, (**b1**) HT-2 and (**c1**) HT-3, and height contour coloring maps in middle column of (**a2**), (**b2**), (**c2**) correspond to crack nucleation zones, and these in right column of (**a3**), (**b3**), (**c3**) correspond to propagation zones.

**Figure 10 materials-15-05528-f010:**
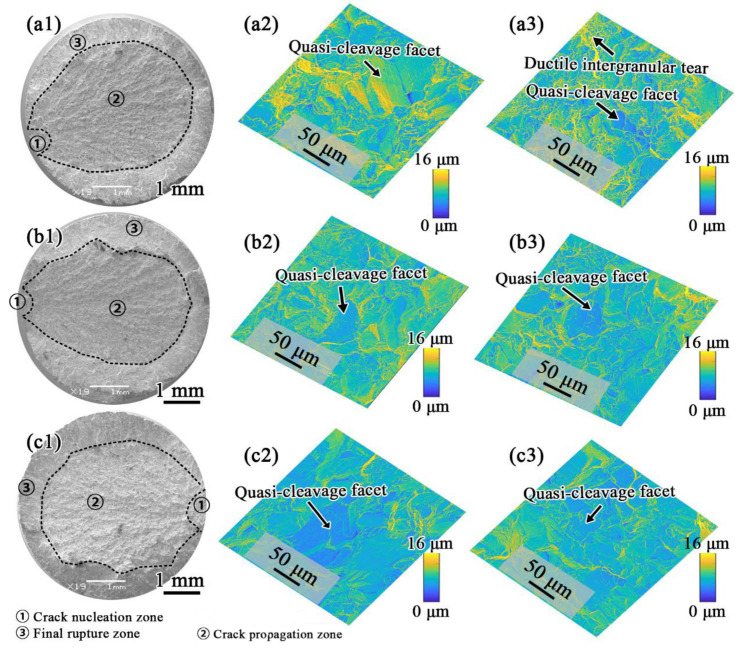
Fractographies of 650 °C tensile specimens of (**a1**) HT-1, (**b1**) HT-2 and (**c1**) HT-3, and height contour coloring maps in middle column of (**a2**), (**b2**), (**c2**) correspond to crack nucleation zones, and these in right column of (**a3**), (**b3**), (**c3**) correspond to propagation zones.

**Figure 11 materials-15-05528-f011:**
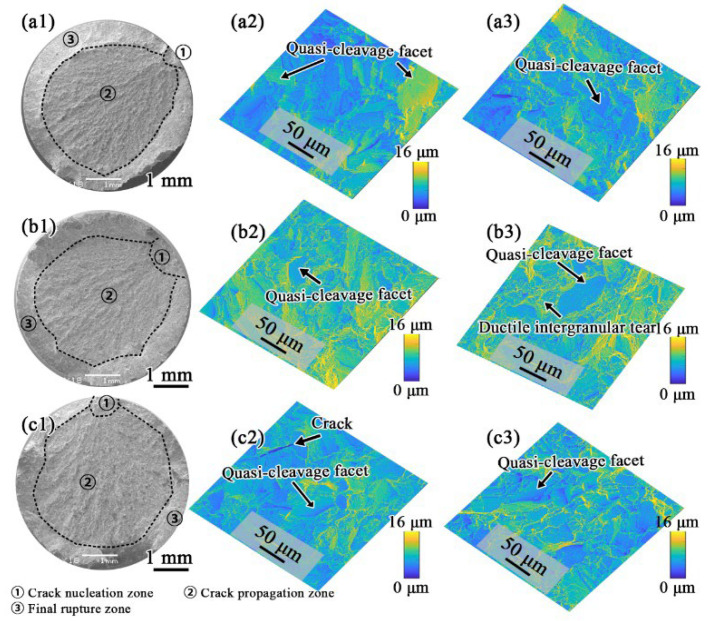
Fractographies of 700 °C tensile specimens of (**a1**) HT-1, (**b1**) HT-2 and (**c1**) HT-3, and height contour coloring maps in middle column of (**a2**), (**b2**), (**c2**) correspond to crack nucleation zones, and these in right column of (**a3**), (**b3**), (**c3**) correspond to propagation zones.

**Figure 12 materials-15-05528-f012:**
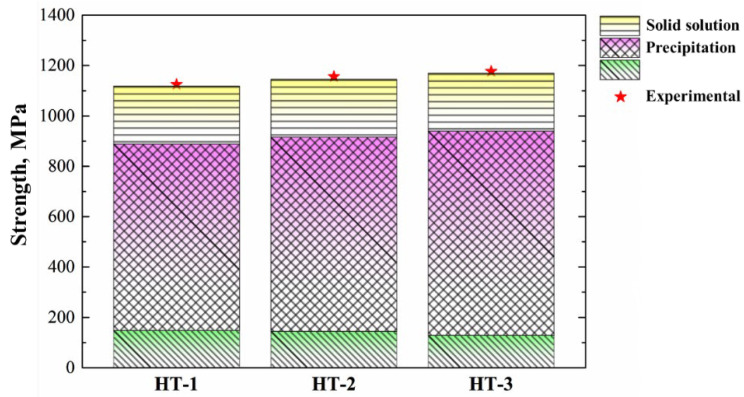
The strength calculated to be contributed by the grain boundary, solid solution strengthening and precipitation strengthening compared to the experimental results for the yield strength at ambient temperature.

**Table 1 materials-15-05528-t001:** The composition of the P/M superalloy FGH4107 in wt.%.

Co	Cr	W	Al	Ti	Mo	Nb	Ta	C	B	Zr	Ni
15.27	10.49	6.30	3.74	4.04	2.88	1.84	0.90	0.036	0.026	0.087	Bal.

**Table 2 materials-15-05528-t002:** The preparation process for the P/M superalloy FGH4107.

	HIP	Solution	Aging
As-HIP	1200 °C/130 MPa/4 h	-	-
HT-1	1200 °C/130 MPa/4 h	1160 °C/1.5 h/SA	871 °C/4 h/AC + 760 °C/16 h/AC
HT-2	1200 °C/130 MPa/4 h	1180 °C/1.5 h/SA	
HT-3	1200 °C/130 MPa/4 h	1200 °C/1.5 h/SA	

**Table 3 materials-15-05528-t003:** The equilibrium compositions in at.% for the γ matrix and γ′ phase at 760 °C in FGH4107, as calculated in JMatPro 6.0.

	Ni	Al	Co	Cr	Mo	Nb	Ta	Ti	W	Zr
γ	41.019	1.566	25.816	27.681	1.718	0.041	0.013	0.186	1.952	0.008
γ′	65.541	12.469	8.864	1.718	0.117	1.874	0.469	8.003	0.889	0.057

**Table 4 materials-15-05528-t004:** *β*_γ_ values of γ matrix elements of Ni-base alloys (MPa/at.%^1/2^) [9,28].

Mo	W	Cr	Co	Nb	Ti	Al	Ta	Zr
101.5	97.7	33.7	39.4	118.3	77.5	22.5	119.1	235.9

**Table 5 materials-15-05528-t005:** Characteristics of γ′ phase in FGH4107.

	Primary γ′		Secondary γ′		Tertiary γ′	
	Volume Fraction, %	Average Size, nm	Volume Fraction, %	Average Size, nm	Volume Fraction, %	Equivalent Diameter, nm
HT-1	5.1	2.8	41.1	280	12.35	67
HT-3	4.3	2.1	39.5	225	14.75	56
HT-5	1.2	0.9	37.6	170	19.75	23

## Data Availability

Data is contained within the article.

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
