# Peer review of "Effects of Solution Temperature on Tensile Properties of a High γ′ Volume Fraction P/M Superalloy"

_materials, 2022, doi:10.3390/ma15165528_

Round 1

Reviewer 1 Report

This paper presents the effect of the temperature of solution treatment in the range of 1160-1200 C on the microstructure and tensile properties and fracture of the nickel based FGH4107 superalloy obtained by powder metallurgy. I would recommend accepting it after some minor revisions. See some comments below:

1.       In the caption of Figure 2 the descriptions for b,c and d should be added.

2.       Equation 4 should be corrected according to the description to this equation: “where τp, pri, τp, sec and τp, ter correspond to critical resolved shear stress (CRSS) of primary, secondary and tertiary γʹ precipitates” (i.e. the τp, ter is used 3 times in the equation).

3.       There is a lack of information about the sizes of γʹ precipitates and their volume fraction in the paper, although the effect of distribution of these precipitates is discussed and these values are used in calculations of precipitation strengthening.

4.       Discussion. The conclusion in discussion section about the role of precipitations in the total strength (the last paragraph) looks like too compressed. It is better to make it more detailed, may be to add the table or graphs with the values of different strengthening factors (from solid solution, grains, particles) and total calculated strength.

5.       Please check the text for grammatical errors. (for ex., page 4, last sentence, “…the size of γ′ precipitates decreases obviously.”)

Reviewer 2 Report

The abstract section  AND  the material and method section must be improved by more points and details. 

Reviewer 3 Report

The present paper includes interesting results regarding the microstructure and tensile properties of the superalloy entitled “Effects of solution temperature on tensile properties of a high 1 γ′ volume fraction P/M superalloy” which can be suitable for Journal of MATERIALS. Anyhow, the reviewer would like to make the following comments

1.       What is the novelty in this article?

2.       The chemical composition and crystallinity of the γ′ phase in the superalloys must be explained in more detail in the introduction section.

3.       Purity and brand name of the raw materials are missing

4.       How many feedstocks were used in this study? Fig.1-1 shows three types of them were used. So, how to mix them?

5.       How did you find the particle size of the pre-alloy powders after VIM?

6.       Based on what standard, was Chemical composition of the super alloy reported in table 1?

7.       The cycle of heat treatment (heating and cooling rate) and concentration of the salt bath are missing in the manuscript.

8.       How did the authors distinguish the primary and send γ′ phase in the Fig.4?

9.       The authors have to justify why with the increasing solution temperature (1160 °C to 1200 °C) ,the size of γ′ precipitates decreases.
